# Can Wastewater Surveillance Enhance Genomic Tracking of Climate-Driven Pathogens?

**DOI:** 10.3390/microorganisms13020294

**Published:** 2025-01-28

**Authors:** Laura A. E. Van Poelvoorde, Erik A. Karlsson, Myrielle Dupont-Rouzeyrol, Nancy H. C. J. Roosens

**Affiliations:** 1Transversal Activities in Applied Genomics, Sciensano, 1050 Brussels, Belgium; laura.vanpoelvoorde@sciensano.be; 2Virology Unit, Institut Pasteur du Cambodge, Phnom Penh 120210, Cambodia; 3URE Dengue et Arboviroses, Institut Pasteur de Nouvelle-Calédonie, Noumea 98845, New Caledonia

**Keywords:** climate change, wastewater surveillance, pathogen detection, public health, infectious diseases

## Abstract

Climate change heightens the threat of infectious diseases in Europe, necessitating innovative surveillance methods. Based on 390 scientific papers, for the first time, this review associates climate-related pathogens, data related to their presence in wastewater, and associated available genomic detection methods. This deep analysis reveals a wide range of pathogens that can be tracked through methods such as quantitative and digital PCR, as well as genomic pathogen enrichment in combination with sequencing and metagenomics. Nevertheless, significant gaps remain in the development of methods, particularly for vector-borne pathogens, and in their general harmonization relating to performance criteria. By offering an overview of recent advancements while identifying critical gaps, we advocate for collaborative research and validation to integrate detection techniques into surveillance frameworks. This will enhance public health resilience against emerging infectious diseases driven by climate change.

## 1. How Will Climate Change Potentially Influence Infectious Diseases in Europe and What Are the Surveillance Priorities?

Climate change is a consequence of the continual release of greenhouse gas emissions into the atmosphere, significantly impacting Earth’s climatic patterns [1,2]. The increase in the frequency of extreme weather events, the changes in precipitation patterns, and the rise in global temperatures are not only environmental challenges but they are also critical public health concerns. It poses a significant threat to human health, including potential alterations to the dynamics of infectious diseases, altering their distribution, frequency, and severity [3]. Changes in temperature, humidity, and rainfall patterns can influence the growth and spread of pathogens, while extreme weather events such as floods, droughts, and heatwaves exacerbate the risks of outbreaks [4,5,6,7,8]. The relationship between infectious diseases and climate change is complex and multifactorial because it involves direct effects on pathogen survival, human vulnerability, and vector distribution [4,5,6,7,8]. Moreover, there are also indirect effects related to societal disruptions, migration, and healthcare infrastructure [9,10]. For example, the societal upheaval and health crises due to the recent COVID-19 pandemic have highlighted the potential consequences of future pathogen outbreaks, bringing renewed attention to this issue.

As seen in Figure 1, three primary categories of infectious diseases can potentially be impacted by climate change: water-borne, food-borne, and vector-borne diseases. Each of these disease categories demonstrates distinct pathways through which climate change impacts disease dynamics, necessitating tailored surveillance and mitigation strategies. Figure 1 shows the prioritization of the pathogens regarding the influence of climate change on the pathogens. First, rising temperatures and more extreme weather patterns contribute to increased floods and altered hydrological cycles, impacting water-borne diseases [4]. Second, changes in weather conditions, such as high temperatures and extreme weather events such as drought and flooding, affect the growth and transmission of food-borne pathogens [11]. Third, alterations in temperature, precipitations patterns, and habitat suitability for disease-carrying vectors impact vector-borne diseases [6,7,8]. Although the link between climate change and the emergence of infectious diseases is widely recognized, the exact degree of human vulnerability to these diseases as a result of climate change remains uncertain [6,11].

Public health is significantly impacted by climate change due to altering hydrological cycles and increasing weather conditions. These heighten the risk of water-borne diseases [4]. The rise in heatwaves leads indirectly to more recreational water activities, consequently leading to a rise in instances of water-borne illnesses such as *Vibrio*-associated infections and other water-borne diseases (Figure 1) [12,13,14]. *Vibrio* bacteria thrive in warm, moderately saline water, and have become more prevalent in rapidly warming seas such as the Baltic and North Seas. An increasing number of days since 1980 have been observed to be suitable for *Vibrio* infections [15,16], with Germany’s coastal regions reporting more cases during hot summers [17]. Moreover, extreme weather events such as storms and floods lead to the displacement of communities and outbreaks of several water-borne diseases such as leptospirosis and others (Figure 1) [1,4,18,19,20]. For example, in Italy, human cases of *Leptospira* occurred in 2009 after heavy rainfall caused street flooding [21]. In New Caledonia, it was shown that during the La Niña phases of the El Niño Southern Oscillation, leptospirosis outbreaks were further exacerbated [22]. Similarly, *Cryptosporidium*, a protozoan parasite causing cryptosporidiosis, is more frequently detected in drinking water following extreme rainfall events, leading to widespread outbreaks [23]. In conclusion, altered temperatures and precipitation patterns impact the distribution and prevalence of water-borne diseases, elevating the risk of inadequate water sanitation and hygiene standards [5,24].

Additionally, food-borne pathogens across the value chain are affected by climate change [11]. Higher temperatures enhance pathogen survival which increase food poisoning incidents, such as *Campylobacter* and *Salmonella*, with particular increases during the warmer months [6,25,26]. Another pathogen of concern is *hepatitis A*, which is often transmitted through the consumption of contaminated shellfish or plants irrigated with polluted water. Due to climate change, there is an increase in floods, which can exacerbate the contamination of water sources, increasing the risk of outbreaks [27]. Additionally, *Shigella*, a pathogen linked to poor sanitation and contaminated water, may see heightened transmission rates as extreme weather events disrupt water quality [28]. Toxoplasma gondii, a protozoan parasite responsible for toxoplasmosis, poses a unique risk in the context of climate change. Their highly resilient oocysts are often transmitted through contaminated water, soil, or food. Warmer climates and altered precipitation patters can facilitate their spread, thus increasing the risk of exposure [29]. Furthermore, contaminated drinking and irrigation water, as well as the breaking down of the cold chain across food processing handling and storage, have been linked to food-borne outbreaks [30,31,32]. These interactions between climate change and the food system complicate the prediction and mitigation of new pathogenic threats, thus emphasizing the need for robust surveillance and adaptive public health strategies.

Vector-borne diseases are emerging due to climate change-induced shifts in vector habitats and activity seasons. Vectors such as mosquitoes, ticks, and sand flies have expanded their habitat ranges and prolonged their activity seasons because of the changes in temperature, precipitation patterns, and habitat suitability [6,7,8]. For instance, *Aedes albopictus*, a vector for at least 22 arboviruses, appeared in 1979 in Europe and has since spread widely, causing dengue outbreaks in Croatia, France, and Italy [33,34]. In 2023, France reported 65 indigenous dengue cases, surpassing the total from 2010 to 2021, including in previously unaffected regions. This spread underscores the urgent need to address vector-borne disease threats [35,36,37]. Another vector-borne disease of concern is malaria, which has the potential of re-emerging in southern Europe due to the expansion of the habitat of the *Anopheles* mosquitoes, particularly during warmer and wetter summers [38]. Additionally, the sandfly-borne disease *visceral leishmaniasis* showed increasing prevalence in Mediterranean countries, with prolonged activity seasons and expanded the habitats of the sandfly [39]. Moreover, climate change has been linked to the spread of *tick-borne encephalitis* (TBE) because the milder winters and earlier springs extend the active period of the *Ixodes* ticks, that carry the TBE virus [40].

Finally, climate change indirectly influences various other diseases such as anthrax, Q fever, tetanus, meningococcal infection, and hantavirus. Rising temperatures, habitat changes, and extreme weather events alter the distribution and transmission of these diseases. These multifaceted health impacts in Europe highlight the necessity for proactive monitoring and mitigation measures to address emerging health risks due to climate change.

According to the classification of Figure 1, we can observe that *Lyme borreliosis*, *Vibrio spp.*, and *Visceral leishmaniasis* are the pathogens that will be affected most by climate change.

**Figure 1 microorganisms-13-00294-f001:**
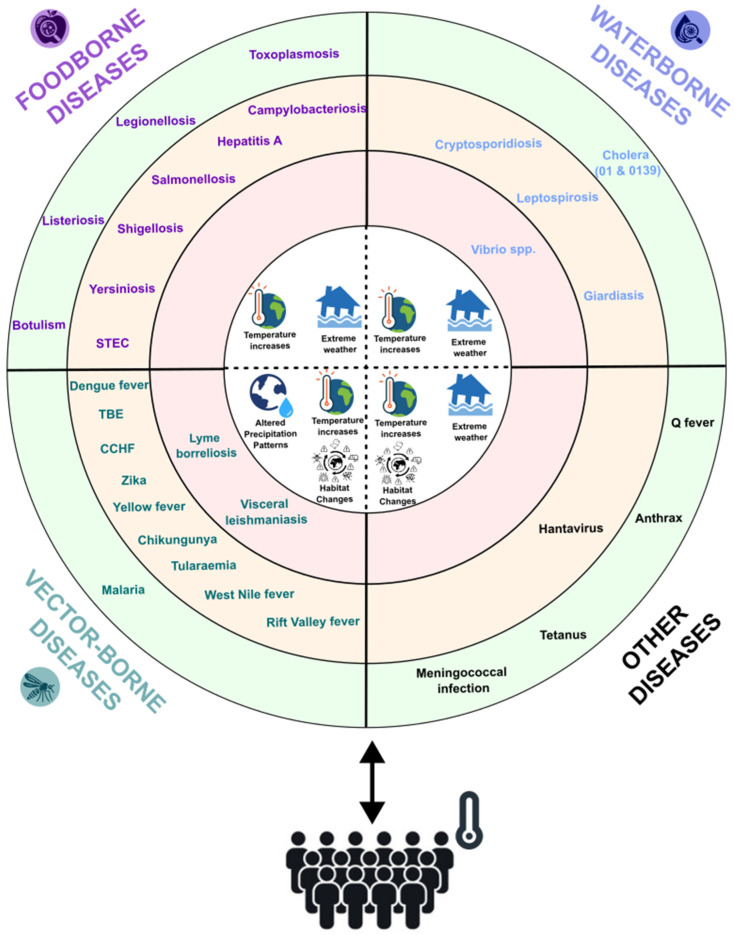
Weighted risk analysis of climate change impacts on infectious disease risks in Europe. Infectious diseases, which impact human health, are influenced by climate change through the triggering of various climatic hazards, such as extreme precipitation, floods, droughts, warming trends, heatwaves, wildfires, and rising sea levels. Infectious diseases are divided based on the strength of influence of climate change on the pathogen (weaker link at the outside of the circle (green); stronger link at the inside of the circle (red)). These pathogens cause disease in humans, which can be detected using, for example, molecular methods. This figure and the ranking of the organisms was constructed based on the data from existing reports and studies [41,42,43]. The Zika virus was included in the figure to account for significant outbreaks that occurred after the table was initially created by Lindgren et al. [41]. It underwent a similar evaluation process as in the original paper. Since then, several large outbreaks of *Zika* have occurred. In August 2019, the first cases of *Zika* virus transmission within Europe were detected in Southern France. Consequently, the impact of the disease on European society was assessed as medium. Regarding its link with climate change, the strength was evaluated as medium. This assessment is based on climate conditions such as high temperatures, flooding, and drought, which promote the spread and abundance of mosquitoes, which are the primary vectors for *Zika* virus transmission. Additionally, water-borne diseases are coloured in blue and are mainly impacted by rising temperatures and extreme weather patterns that contribute to increased floods [4]. Food-borne diseases are coloured in purple and are influenced by changes in weather conditions such as rising temperatures and extreme weather [11]. Finally, vector-borne diseases are highlighted in teal and are impacted by alterations in temperature, precipitation patterns, and habitat suitability for disease-carrying vectors. The other diseases are influenced by rising temperatures, extreme weather patterns that contribute to increased floods, and habitat changes [6,7,8] (CCHF = Crimean–Congo haemorrhagic fever; STEC = Shiga toxin-producing E. coli; TBE = tick-borne encephalitis).

## 2. What Advancements Should Be Considered to Enhance Wastewater Surveillance for Public Health?

Wastewater surveillance has gained traction as a proactive and effective methodology with immense potential in redefining disease surveillance frameworks. It contributes to a more comprehensive, community-level perspective on disease dynamics and potential threats, adding an additional layer to early warning systems. However, wastewater surveillance needs to be integrated with epidemiological and clinical information to obtain a comprehensive understanding of disease dynamics and risk. Addressing this concern, the Urban Wastewater Treatment Directive [44] mandates EU Member States to monitor specific public health parameters in urban wastewater, including emerging pathogens associated with climate change.

Innovative approaches are of critical importance for the early detection and monitoring of infectious diseases, especially due to emerging threats associated with climate change. In addition, pathogen detection and analysis tools have advanced significantly over time. The most commons molecular methods are Quantitative and Droplet Digital Polymerase Chain Reaction (qPCR and ddPCR), genomic pathogen enrichment and sequencing, and metagenomics (Figure 2). Each tool offers unique advantages and disadvantages, which are explored in detail below. Validated (RT-)qPCR or (RT-)ddPCR methods allow for the rapid, efficient, and precise quantification of viral loads within wastewater [45,46,47,48,49]. Moreover, these methods are highly sensitive and specific. They only require a minimal nucleic acid input to detect the targeted pathogen, which is particularly interesting for early-stage disease detection [50]. Moreover, it makes them accessible for routine surveillance. Along with epidemiological modelling, these methods allow the monitoring of disease incidence in the population. Normalization by population parameters or other indicators can further refine assessments of contamination levels within the community. However, these methods rely on predefined targets, potentially missing novel or unknown pathogens or mutations. This highlights the need for complementary approaches to ensure comprehensive surveillance.

Furthermore, genomic pathogen enrichment and sequencing techniques enable phylogenetics analysis, evolutionary studies, and functional characterization of mutations or variations within the targeted pathogen. Genomic pathogen enrichment encompasses methods such as probe capture methods, amplicon generation, unbiased enrichment, host depletion, adaptive sampling, and unbiased amplification, each with distinct advantages and limitations. Probe capture methods use specific oligonucleotide probes that bind to the target sequences, which selectively enriches them from a complex mixture of nucleic acids [51]. For instance, probe capture methods have been used to detect viral pathogens in wastewater, such as *norovirus*, *rotavirus A*, *hepatitis A* virus, and *RSV*. Moreover, studies that use these methods have revealed variations in virus concentrations across different cities, reflecting the local transmission of the pathogens [52]. Seasonal fluctuations were also observed with higher concentrations of certain viruses, such as influenza, often detected during colder months, reflecting the seasonal epidemic. Although this method is highly specific and can efficiently target particular pathogens or genomic regions, its specificity is a limitation because it may exclude unknown or poorly characterized sequences that do not match the probe design. Amplicon generation relies on the PCR amplification of specific genomic regions. This method is particularly effective in the detection and characterization of known pathogens or mutations because it allows us to focus on regions of interest with high sensitivity and specificity [51]. This method has been used regularly in the characterization of SARS-CoV-2 in wastewater. The different SARS-CoV-2 variants and their dynamics could be observed in wastewater and aligned closely to the profile reported by clinical sequencing [53]. However, due to primer design and PCR conditions, amplicon generation can introduce biases, which potentially skews the results. Moreover, the detection of sequences outside the targeted regions may fail. Host depletion aims to remove host-derived nucleic acids and, consequently, the microbial content within a sample will be enriched, which can enhance the detection of low-abundance pathogens by reducing background noise [54]. However, this approach has some risks, such as the unintended removal of target sequences. Moreover, the efficiency of host depletion can vary depending on the method and sample composition, which can lead to inconsistencies in results. Finally, adaptive sampling is an emerging technique facilitated by nanopore sequencing technology. This real-time approach dynamically adjusts sequencing efforts to focus on underrepresented regions or targets within a sample. One of the advantages is the flexibility, allowing researchers to prioritize specific sequences while data are generated. However, it is still in its developmental stage and requires sophisticated computational algorithms and hardware. Unbiased amplification methods aim to amplify the entire genome without favouring specific regions. These methods are advantageous when the target is unknown or when a comprehensive genomic representation is required. Nevertheless, this technique can introduce amplification biases, leading to uneven coverage and the potential overrepresentation of certain sequences. All of these methods may be biased towards certain pathogens or variants, and there can be challenges in obtaining sufficient sequence coverage for accurate analysis. Moreover, these methods often require a higher nucleic acid input of the targeted pathogen than qPCR and ddPCR methods.

Finally, unbiased metagenomics offers the possibility of detecting a broad spectrum of pathogens within a single sample [55]. Unlike targeted methods, unbiased metagenomics analyzes all nucleic acids present in a sample, including RNA and DNA from viruses, bacteria, fungi, and other organisms. This approach can detect both known and unknown pathogens, enabling the identification of emerging and poorly characterized microorganisms that may not be captured by targeted assays. Moreover, this ability enhances the response to both old and new disease threats, providing critical insights into pathogen diversity and dynamics in complex matrices such as wastewater [52,56,57]. Unlike targeted methods, metagenomics does not rely on predefined sequences, making it invaluable in helping to understand the diversity and co-occurrence of strains and pathogens, and potentially identify new or emerging pathogens. However, it is worth noting that metagenomics is still heavily research-oriented with numerous tools and methods yet to be optimized, making achieving standardization challenging. Unlike qPCR and ddPCR methods, which can potentially be harmonized with relative ease given the right incentives and resources, metagenomics presents greater challenges in standardization. Moreover, metagenomics needs a high nucleic acid input of the targeted pathogen and may suffer from challenges in distinguishing pathogens from background noise and difficulties in interpreting complex generated data. Sensitivity issues may arise when attempting to detect a pathogen with a small genome or one that is present in low abundance within a sample dominated by nucleic acids from other organisms. For example, in wastewater, viral genomes such as the *norovirus*, *influenza virus*, *rotavirus*, may be present at very low concentrations within the sample, making their detection challenging in the presence of an overwhelming amount of bacterial DNA or eukaryotic nucleic acids or contaminants. Additionally, metagenomics requires substantial computational resources and expertise for data analysis. There can also be variability in results depending on the bioinformatics pipeline used, which highlights the urgent need for the standardization of metagenomic workflows. Further exploration into the possibilities of using these tools to observe and comprehend the dynamics of climate change and pathogens in wastewater is warranted.

Finally, after detection and analysis, clear and timely reporting is necessary to communicate findings from wastewater surveillance to stakeholders, including public health authorities and the general public. Real-time data sharing platforms can help alert health officials to emerging threats, enabling rapid responses to contain outbreaks. Standardized reporting formats are also critical for comparing data across regions, ensuring consistency in surveillance efforts and aligning with clinical findings for validation and action.

## 3. Can Wastewater Surveillance Be an Effective Tool for Tracking Climate Change-Related Pathogens?

Wastewater surveillance shows promise as a tool for monitoring and detecting a wide array of pathogens. In various countries, wastewater surveillance has already been successfully integrated into public health systems, demonstrating its utility as an early warning system for disease outbreaks and as a tool for ongoing public health monitoring. For instance, during the COVID-19 pandemic, wastewater surveillance was rapidly introduced in countries such as Belgium, Australia, and the United States. In Belgium, wastewater monitoring for SARS-CoV-2 RNA provided critical data on infection trends at the community level. Moreover, the use of wastewater surveillance in combination with genomic sequencing enabled the health authorities to trace specific viral strains. Additionally, the Belgian wastewater surveillance program has expanded to the monitoring of other respiratory viruses such as RSV and seasonal influenza [58]. Another example of the use of wastewater surveillance is by the World Health Organization (WHO) South East Asia Region that monitors the poliovirus in sewage samples to monitor the effectiveness of containment activities and detect any breach in containment [59]. In Finland, the wastewater surveillance system is integrated into its national public health framework to monitor the prevalence of illicit drug use. This showcases the versatility of wastewater surveillance beyond pathogen detection [60]. Additionally to these examples, wastewater surveillance can be used to detect pathogens linked to climate change, with various detection methods already being utilized. A deep analysis was carried out to associate climate-related pathogens, their presence in wastewater, and the available advanced detection methods (see Table 1).

In Table 1, it is shown that all water-borne and food-borne pathogens associated with climate change were detected in wastewater. However, the presence of most vector-borne pathogens in wastewater has not (yet) been confirmed (Table 1). Nevertheless, all the vector-borne pathogens were detected in either urine or feces (Table 1), indicating future potential detectability in wastewater samples. Therefore, it can be concluded that wastewater surveillance has the potential to track the dynamics of most pathogens related to climate change, as the majority of these microorganisms (29/31) are present or suspected to be present in wastewater. The exceptions are *Bacillus anthracis* and *Francisella tularensis*, which have not been detected in either urine or feces. These two pathogens are considered to have a lower impact from climate change, as shown in Figure 1.

Regarding the detection methods tested in wastewater, the analysis reveals that a range of methods, from the classical culturing or isolation of the pathogens to the most advanced ones like metagenomics, have been used. Some pathogens may benefit from isolation or culture before employing molecular detection methods, depending on the specific requirements of the detection process. Isolation or culture can facilitate the detection process, especially if it aligns with the current gold standard for detection (Appendix A). Bypassing the need for prior isolation enables a more efficient and less time-consuming detection of pathogens. Consequently, this enhances the effectiveness of wastewater surveillance as a robust tool in disease monitoring and public health risk assessment.

The study also reveals significant gaps, as well as the need to develop specific detection methods, because for eleven pathogens, no method was described for their detection in wastewater although they have the potential to be present. This is especially critical for vector-borne pathogens, many of which are categorized as having a medium to high climate change impact (Figure 1).

Another challenge lies in the lack of harmonization and reliability in pathogen detection protocols. The lack of harmonization and reliability in pathogen detection protocols is particularly critical for global monitoring efforts and for making meaningful comparisons between European countries. While PCR and sequencing methods are commonly used, they often lack standardization, which complicates global monitoring efforts and cross-country comparisons. Metagenomics, despite its potential, has been underused in pathogen detection for wastewater surveillance and when used, it faces challenges related to protocol optimization, RNA degradation, and low viral loads. The further development and optimization of detection methods are essential for wastewater surveillance to reach its full potential. Moreover, achieving full harmonization also requires a substantial investment in training, infrastructure, and collaborative frameworks. Cross-border collaboration is further hindered by disparities in technical capacity and funding, emphasizing the need for equitable resource distribution and knowledge sharing.

## 4. Concluding Remarks and Future Perspectives

Escalating climate change leads to an intensified impact of climatic hazards, which represents an imminent threat to public health through the intensification and emergence of infectious diseases in Europe. Wastewater surveillance presents both opportunities and challenges for its effective integration into disease surveillance frameworks. By collecting the available scientific data about pathogen detection in wastewater, this study highlights the crucial role of surveillance in detecting and monitoring infectious diseases linked to climate change in Europe. The EU’s Urban Wastewater Treatment Directive mandates monitoring specific health parameters, aligning with the goal of tracking climate-related pathogens.

Most pathogens linked to climate change can be detected in wastewater or are suspected to be present based on findings in urine or feces. Molecular methods like culturing, PCR-based techniques such as qPCR and ddPCR, and enrichment sequencing are commonly used for routine pathogen detection in wastewater surveillance; however, there is a gap concerning some of the climate change-related pathogens, especially vector-borne viruses. For these pathogens, it is important that specific qPCR or ddPCR methods are tested, and a specific sequencing or metagenomics workflow is developed. A major challenge is standardizing methodologies for consistent pathogen detection, especially for cross-country surveillance systems, as proposed by the EU Directive.

This paper represents an innovative study, linking climate-related pathogens to wastewater detection methods, offering a novel framework for future surveillance strategies. Future efforts should focus on developing validated methods to detect and characterize pathogens in wastewater. This includes the improvement of the sensitivity and specificity of the methods, developing cost-effective workflows, and addressing challenges related to nucleic acid degradation and low pathogen concentrations in wastewater. Advancing sequencing and metagenomics techniques to detect pathogens is essential to address climate change-related challenges, but high costs and complexity may limit accessibility, particularly in resource-limited settings. These resource-limited regions, in particular, require equitable access to these tools to ensure that surveillance efforts are globally inclusive and effective. Overcoming these challenges requires developing cost-effective, standardized metagenomics techniques tailored to wastewater surveillance. Moreover, international funding mechanisms and capacity-building initiatives should be established to bridge these gaps, enabling a broader participation in wastewater surveillance networks.

There is a pressing need for continued research and the integration of wastewater surveillance into monitoring and response activities. Effective wastewater surveillance requires strengthened surveillance infrastructure, developing standardized protocols, and enhancing interdisciplinary collaboration among researchers, public health authorities, and policy makers. They must work together to align priorities, share expertise, and foster innovation. Moreover, these collaborations should extend to the development of global databases and repositories for wastewater-derived pathogen data. Additionally, interdisciplinary studies should be performed to explore the links between wastewater data, epidemiological trends, and environmental factors. This will enable a comprehensive analysis of trends and threats at an international level, allowing for an understanding of disease dynamics in the context of climate change. This approach enables timely responses to emerging infectious disease threats, safeguarding public health in the face of climate change while setting a new standard for proactive health surveillance.

In conclusion, the transformation of wastewater into a critical tool for public health monitoring presents an unparalleled opportunity to address the dual threats of climate change and emerging infectious diseases. By strengthening wastewater surveillance systems, standardizing methodologies, and fostering international collaboration, it will advance our ability to predict, detect, and respond to these interconnected challenges, ultimately safeguarding global health in an era of unprecedented environmental change.

## Figures and Tables

**Figure 2 microorganisms-13-00294-f002:**
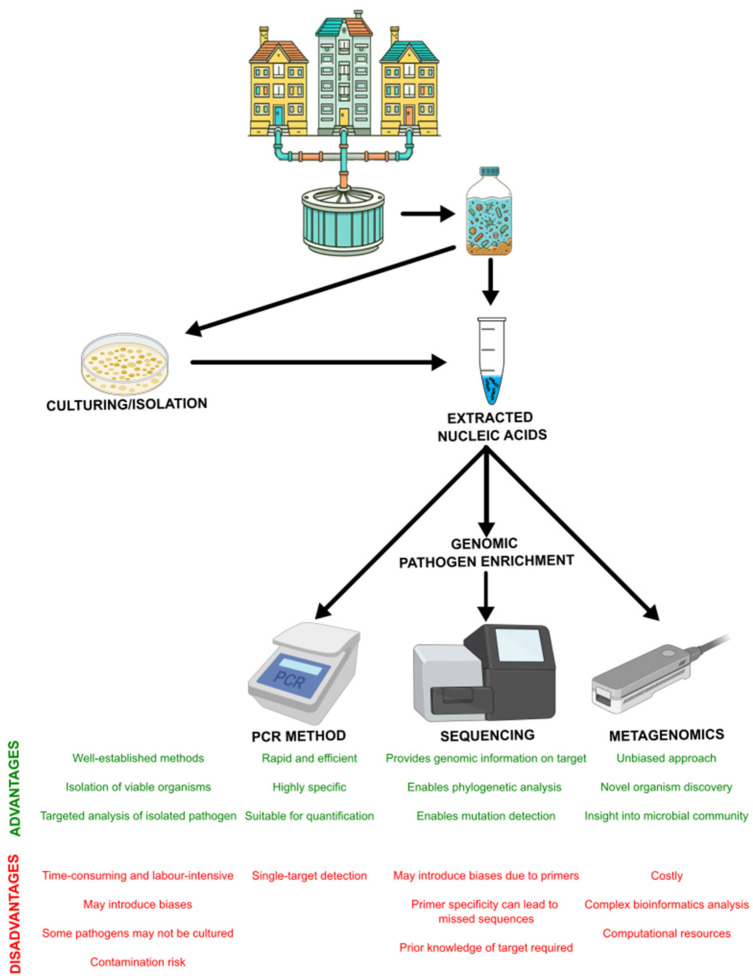
Wastewater surveillance pathogen detection. A person infected with a pathogen can excrete pathogens and their genomes via wastewater. A wastewater sample can be collected for molecular analysis, such as PCR methods (with a prior culturing or isolation step), sequencing (with a prior culturing or isolation step or genomic pathogen enrichment step), and metagenomics. The integration of these advanced techniques enhances the effectiveness of wastewater surveillance in providing early warnings and assessing public health risks.

**Table 1 microorganisms-13-00294-t001:** Summary of pathogen detection in urine, feces, and wastewater using various molecular methods. The table provides an overview of all pathogens from Figure 1 that were reported in published literature to be detected in urine, feces, and/or wastewater samples, along with their employed methodologies. These methods include prior culturing or isolation followed by a molecular method such as PCR methods or next-generation sequencing (NGS) and whole-genome sequencing (WGS). The use of PCR methods and genomic pathogen enrichment methods, including (nested) PCR and amplicon sequencing, was employed in combination with sequencing and metagenomics. Detection in urine and/or feces was included because while some of these pathogens have not (yet) been found in wastewater, their detectability in urine or feces indicates their potential to be found in wastewater. The font of water-borne diseases is blue, the font of food-borne diseases is green, and the font of vector-borne diseases is purple. Moreover, the cells of the pathogens are coloured according to the potential influence of climate change on the pathogens (low = green; medium = orange; high = red). (CCHF = *Crimean–Congo haemorrhagic fever*; STEC = Shiga toxin-producing *E. coli*; TBE = tick-borne encephalitis). If published literature was available that confirms the possible detection of particular pathogens in urine, feces, or wastewater, this is indicated by “+”. In Appendix A, references to the literature can be found. If no published literature was available that confirms the possible detection of particular pathogens in urine, feces, or wastewater, this is indicated by “−”.

Targets	Confirmed Presence in …			Detection of Pathogens in Wastewater with Following Methods			
	Urine	Faeces	Wastewater	Prior Culturing or Isolation + Molecular Method	PCR Method	Genomic Pathogen Enrichment Methods + Sequencing	Metagenomics
*Cryptosporidium*	−	**+**	**+**	−	**+**	**+**	**+**
*Giardia*	−	**+**	**+**	−	**+**	**+**	**+**
*Leptospira*	**+**	−	**+**	**+**	−	**+**	−
** *Vibrio* **	−	**+**	**+**	**+**	**+**	**+**	**+**
*Clostridium botulinum*	**+**	**+**	**+**	**+**	−	−	−
*Toxoplasma*	**+**	−	**+**	−	**+**	**+**	**+**
*Listeria*	**+**	**+**	**+**	−	−	−	**+**
*Campylobacter*	−	**+**	**+**	**+**	**+**	**+**	**+**
*Hepatitis A*	**+**	**+**	**+**	**+**	**+**	**+**	**+**
*Legionella*	**+**	−	**+**	**+**	**+**	**+**	**+**
*Salmonella*	**+**	**+**	**+**	**+**	**+**	**+**	**+**
*Shigella*	**+**	**+**	**+**	**+**	**+**	**+**	**+**
*STEC*	**+**	**+**	**+**	**+**	−	−	−
*Yersinia*	**+**	**+**	**+**	−	**+**	**+**	**+**
*Plasmodium (Malaria)*	**+**	**+**	−	−	−	−	−
*CCHF*	**+**	−	−	−	−	−	−
*Chikungunya virus*	**+**	−	**+**	−	**+**	−	**+**
*Dengue virus*	**+**	−	**+**	−	**+**	−	−
*Rift Valley fever*	**+**	**+**	−	−	−	−	−
*TBE*	**+**	−	−	−	−	−	−
*West Nile fever*	**+**	−	−	−	**+**	−	−
*Yellow Fever*	**+**	−	−	−	−	−	−
*Zika virus*	**+**	−	**+**	−	**+**	−	−
** *Borrelia (Lyme disease)* **	**+**	−	−	−	−	−	−
** *Leishmania (Visceral Leishmaniasis)* **	**+**	−	−	−	−	−	−
*Bacillus anthracis (Anthrax)*	−	−	−	−	−	−	−
*Neisseria meningitidis (Meningococcal infection)*	**+**	−	−	−	−	−	−
*Coxiella Burnetii (Q fever)*	**+**	**+**	**+**	−	**+**	−	−
*Clostridium tetani (Tetanus)*	−	**+**	−	−	−	−	−
*Francisella tularensis (Tularaemia)*	−	−	−	−	−	−	−
*Hantavirus*	**+**	−	−	−	−	−	−

## Data Availability

No new data were created or analyzed in this study. Data sharing is not applicable to this article.

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
