# Peer review of "Can Wastewater Surveillance Enhance Genomic Tracking of Climate-Driven Pathogens?"

_microorganisms, 2025, doi:10.3390/microorganisms13020294_

Round 1
Reviewer 1 Report
Comments and Suggestions for Authors
The review article “From qPCR to Metagenomics: Advancing Wastewater Surveillance for Climate-Related Pathogen Detection” summarizes the tools needed to quantify pathogen presence influenced by climate change. Overall, the paper is focused on a summary of the advantages and disadvantages of current molecular tools.
Please consider the following major comments to improve the article.
1) The abstract (line 16) does not clarify what a balanced account means. How did the authors provide a balanced accounting of recent advancements? Further, the article spends a lot of time focused on classical approaches to pathogen detection.
2) Figure 1 does not match the first summary text. For example, the text highlights three climate-related phenomena that can explain the growing presence of pathogens, including 1) increased precipitation and flooding, 2) increases in temperature, and 3) changes in habitats that result in the expansion of vector-based diseases. However, Figure 1 shows diseases based on the mode of transmission. I think Figure 1 would benefit from clarity and the inclusion of the climate change mechanisms influencing the particular pathogens. Further, the other category needs a better explanation.
3) The ranking of organisms in Figure 1 requires justification. How were organisms placed in areas of importance? Also, I’d recommend putting pathogens more influenced by climate change towards the center of the figure (more influence = closer to the inner circle labeled “Climate Change.”
4) Figure 1 caption requires some clarification. What 2011 table was used as a basis for the figure (please include the specific citation)?
5) The section entitled “Rethinking Disease Surveillance through Wastewater…” does not seem appropriately titled for the section. The section is mostly focused on the types of measurements, not Europe’s Health Strategy.
6) Line 122 mentions novel sample types and collection strategies, which are not discussed in the paper. Are these important to consider? Similarly, in line 128, the authors mention modeling but provide no meaningful discussion on the topic.
7) For Figure 2 (and the associated text), it seems a better way to classify the detection is based on direct sample measurement or an initial sample manipulation to enrich the pathogen (either through culturing or pathogen enrichment/host depletion). Next, this would follow with the different types of measurements, including PCR and sequencing. In Figure 2, why is there a separate toilet in the image that doesn’t connect to a sewer system? Does this indicate the difference between centralized and decentralized wastewater treatment systems? Clarity is recommended.
8) If a comparison of techniques is a motivating factor of this review article, I think a deeper, more meaningful comparison of the approaches is required.
9) A significant portion of the section “Can Wastewater Surveillance Be An Effective Tool for Tracking…” repeats similar information from previous sections. Consider reorganizing to collocate information better.
Consider these minor comments
1) Several sections are written as one long paragraph with too many ideas. Ideas should be organized and presented in paragraphs of appropriate size.
2) Microbial nomenclature should be followed. Names should be italicized where appropriate.
3) Line 139, revise redundant phrasing “amplicon amplification.”
4) Line 131, fix grammar error (delete “be”).
5) The supplemental table should have a designation of Table S1 (or other naming mechanisms) to differentiate it from Table 1 in the main paper.
Comments on the Quality of English Language
Overall, spelling and grammar was fine (a few spelling errors). Sections were a bit repetitive, and information could be condensed.
Reviewer 2 Report
Comments and Suggestions for Authors
The manuscript entitled “From qPCR to Metagenomics: Advancing Wastewater Surveillance for Climate-Related Pathogen Detection” associates for the first time climate-related pathogens, data related to their presence in wastewater and associated available genomic detection methods. The work is very interesting, but I would like the author to address my questions before its acceptance.
1. In the abstract, the author said they were based on 388 scientific papers. Do all the references listed in the present work?
2. In line 13, why does the author mention the digital PCR? There is no mention in the title.
3. English needs to be improved. For example, in line 42, ‘Public health is significantly impact by climate change…’. In line 47, ‘Vibrio bacteria, thriving in warm, …’. In line 52, some space needs to be removed. In line 85, the ‘vibrio’ should be italic.
4. What are the meanings of the different circles in Figure 1? Please add the statements in the legends.
5. In line 125, what does the author mean by pathogen enrichment and sequencing?
6. What concentration of the collected sample or DNA would be used for further analysis?
Comments on the Quality of English Languageneed to be improved.
Round 2
Reviewer 2 Report
Comments and Suggestions for Authors
no comments
Comments on the Quality of English Languageno comments